# Computational screening of potential glioma-related genes and drugs based on analysis of GEO dataset and text mining

Zhengye Jiang[1,2], Yanxi Shi[3], Guowei Tan[1,2], Zhanxiang Wang[1,2]*

**1** Department of Neurosurgery, Xiamen Key Laboratory of Brain Center, the First Affiliated Hospital of Xiamen University, Xiamen, China, **2** Institute of Neurosurgery, School of Medicine, Xiamen University, Xiamen, China, **3** Department of Cardiology, Jiaxing Second Hospital, Jiaxing, China

* wzx3819@163.com

## Abstract

### Background

Considering the high invasiveness and mortality of glioma as well as the unclear key genes and signaling pathways involved in the development of gliomas, there is a strong need to find potential gene biomarkers and available drugs.

### Methods

Eight glioma samples and twelve control samples were analyzed on the GSE31095 datasets, and differentially expressed genes (DEGs) were obtained via the R software. The related glioma genes were further acquired from the text mining. Additionally, Venny program was used to screen out the common genes of the two gene sets and DAVID analysis was used to conduct the corresponding gene ontology analysis and cell signal pathway enrichment. We also constructed the protein interaction network of common genes through STRING, and selected the important modules for further drug-gene analysis. The existing antitumor drugs that targeted these module genes were screened to explore their efficacy in glioma treatment.

### Results

The gene set obtained from text mining was intersected with the previously obtained DEGs, and 128 common genes were obtained. Through the functional enrichment analysis of the identified 128 DEGs, a hub gene module containing 25 genes was obtained. Combined with the functional terms in GSE109857 dataset, some overlap of the enriched function terms are both in GSE31095 and GSE109857. Finally, 4 antitumor drugs were identified through drug-gene interaction analysis.

### Conclusions

In this study, we identified that two potential genes and their corresponding four antitumor agents could be used as targets and drugs for glioma exploration.

**Data Availability Statement:** All GSE files are available from the https://www.ncbi.nlm.nih.gov/geo/query/acc.cgi?acc=GSE31095 database.

**Funding:** This study was funded by the National Natural Science Foundation of China, grant number 82072777 to ZW. The funders had no role in study design, data collection and analysis, decision to publish, or preparation of the manuscript.

**Competing interests:** The authors have declared that no competing interests exist.

## Introduction

Glioma is not only a very high degree of malignancy, but also a primary brain tumor with a high recurrence rate and poor prognosis, with an incidence of 3.19 cases per 100,000 person years [1]. Although some progress has been made in early diagnosis, the majority of patients are still at an advanced stage of diagnosis, resulting in extremely high rates of mortality and disability in these patients [2]. According to current medical treatment standards, even with the maximum safe resection, the rate of early recurrence after surgery is extremely high due to the inherent ability of tumor cells to infiltrate the normal brain [3]. Besides, the average overall survival time (OS) of GBM patients is only 12–18 months even after the combination of external irradiation and temozolomide combined with (TMZ) and maintenance chemotherapy, [4,5]. At present, given that gliomas are prone to relapse after treatment and have an inferior prognosis, it is necessary to strengthen the research on the pathogenesis of glioma and explore the genetic markers of glioma, so as to provide the diagnosis and treatment basis for early clinical screening and treatment.

Over the past few years, molecular diagnostics, drug target discovery and other techniques that analyze differences in gene expression have become a hot topic in clinical cancer research. A public database supported by the National Center for Biotechnology Information (NCBI), the Comprehensive Gene Expression Database (GEO), contains dozens of basic disease gene expression profile in the experiment. Currently, GEO databases are being used extensively to identify and mine key genes and underlying mechanisms involved in disease progression [6]. Text mining of biomedical literature has been recognized as a reliable hypothesis-generating method that can reveal novel associations between genes and disease occurrence [7,8]. Although a great deal of research has been carried out on glioma in recent years, the specific pathogenesis of glioma remains unclear. Therefore, we combine gene expression chips with text mining, and analyze these data through modern approach software to find clinically meaningful clues, so as to gain new perspectives, such as new diagnostic gene markers and therapeutic targets [9,10].

In this article, we downloaded the GSE31095 gene expression datasets, which included eight glioma samples and twelve normal controls, from the Gene Expression Omnibus database (GEO) and identified differentially expressed genes (DEGs) by R software (version 3.6.3) [11,12]. Meanwhile, all the glioma genes were mined from the text mining. The intersection of the gene sets obtained from DEG and text mining was analyzed via the online tool Venny to obtain the common genes, and different bioinformatics methods were further used to conduct gene ontology, signaling pathway enrichment annotation, and protein and protein interaction research on these common genes. We then validated our results on another independent GSE109857 dataset. From these data, we could find the gene markers and related pathways that might be associated with glioma, which providing new insights into the molecular mechanism of hidden gliomas.

## Methods

### Data collection

We abstracted the gene expression chip data GSE31095 [13] and GSE109857 from the NCBI Gene Expression Comprehensive (GEO) web resource (https://www.ncbi.nlm.nih.gov/geo/) [6,14]. The GSE31095 cohort contains eight glioma samples and twelve normal control samples, while the GSE109857 dataset includes five glioma samples and five normal control samples.

### Data preprocessing

The core R package was used to process the downloaded matrix files. After normalization, the differences between glioma and the control group were determined by truncation criteria

(|log2 fold change (FC)| $\geq$ 2, FDR < 0.05), and selected the remarkable DEGs for downstream analyses [14,15].

## Text mining

Text mining was based on web services GenCLIP3 platform (http://ci.smu.edu.cn/genclip3/analysis.php/). When manipulated, GenCLIP3 was further used to retrieve all the gene names found in the existing literature relevant to the search topic [16]. We searched for the concept of glioma and screened all the genes associated with the topic from the results. The gene set obtained by text mining further intersected with the previously obtained differential gene set for the next step of analysis.

## Gene ontology analysis and KEGG pathway analysis

To characterize Gene products and their functional characteristics, we used a Gene ontology (GO) approach and provided a standard vocabulary for corresponding terms. The GO terms included biological processes (BP), cellular composition (CC), and molecular function (MF), which reflected the current understanding of genes [17,18]. The Kyoto Encyclopedia of Genes and Genomes (KEGG) database, as an open access informatic source for explaining the biological functions of organic systems, provides a large number of known biological pathways data resources, and the resources are comments for with their respective KEGG pathway of a gene or group of genes/proteins. Besides, a variety of online tools for functional and path enrichment analysis were further used to interpret the resulting intersection function and signal path analysis [19]. FDR<0.05 was considered to be statistically significant.

## PPI network and module analysis

The resulting common set of genes obtained from the online database STRING, a database of 3.1 billion interactions across about 5 K organisms [20], was uploaded to the database for retrieving interacting genes [21]. Steps were as follows. The list of selected genes was firstly mapped to the STRING site to evaluate their interactions. And the genes were selected, when the PPIs comprehensive score was >0.9 and the degree of close correlation with other genes was adjusted to $\geq$10 [22]. After selected, the genes were constructed into a PPI network using Cytoscape visualization software [23]. MCODE was further used to classify the vital gene modules, and the related parameter standards were set by default, except k-core = 7. The genes of the selected module were finally analyzed by functional enrichment with FDR< 0.05 as the standard.

## Drug-gene interaction and functional analysis of potential genes

Through drug-gene interaction, the obtained glioma genes were combined with existing drugs to analyze and explore the potential targets of glioma. Drug gene interactions database (DGIdb: https://www.dgidb.org) is an open-source web site for browsing and filtering drug-gene interactions [24]. As potential therapeutic targets, the module genes were uploaded to the drug-gene database to be match with the existing drugs to obtain the potential genes that match the drugs.

## Results

### DEGs identification and Text mining

Firstly, 463 DEGs were selected from glioma samples and normal controls in the GSE31095 dataset through limma package built-in R software. Then 424 upregulated genes and 39

downregulated genes were selected. Meanwhile, 528 differentially expressed genes, including 186 upregulated genes and 342 downregulated genes, were obtained by analyzing the giloma samples in the GSE109857 dataset and the normal control group. The criteria were set|log2 fold change (FC)|≥ 2 and adjusted $P$ <0.05.

Through text mining, 4155 human genes associated with glioma. After the DEGs in the microarray data were crossed, the intersection of selected genes was obtained, and 128 genes involved in GSE31095 and 127 genes involved in GSE109857 were obtained (Fig 1).

## Function and signal pathway enrichment analysis

To establish the potential roles of the GSE31095 dataset DEGs, we carried out GO term analysis on the 463 genes. GO term analysis indicated that these genes were enriched for immune response (BP), inflammatory response (CC), and plasma membrane and receptor activity (MF) (Fig 2A), respectively. KEGG pathway analysis revealed 13 significantly enriched pathways. The top-5 most enriched pathways were: Tuberculosis, RNA transport, NF-kappa B signaling pathway, Hematopoietic cell lineage, and Natural killer cell-mediated cytotoxicity (Fig 2B).

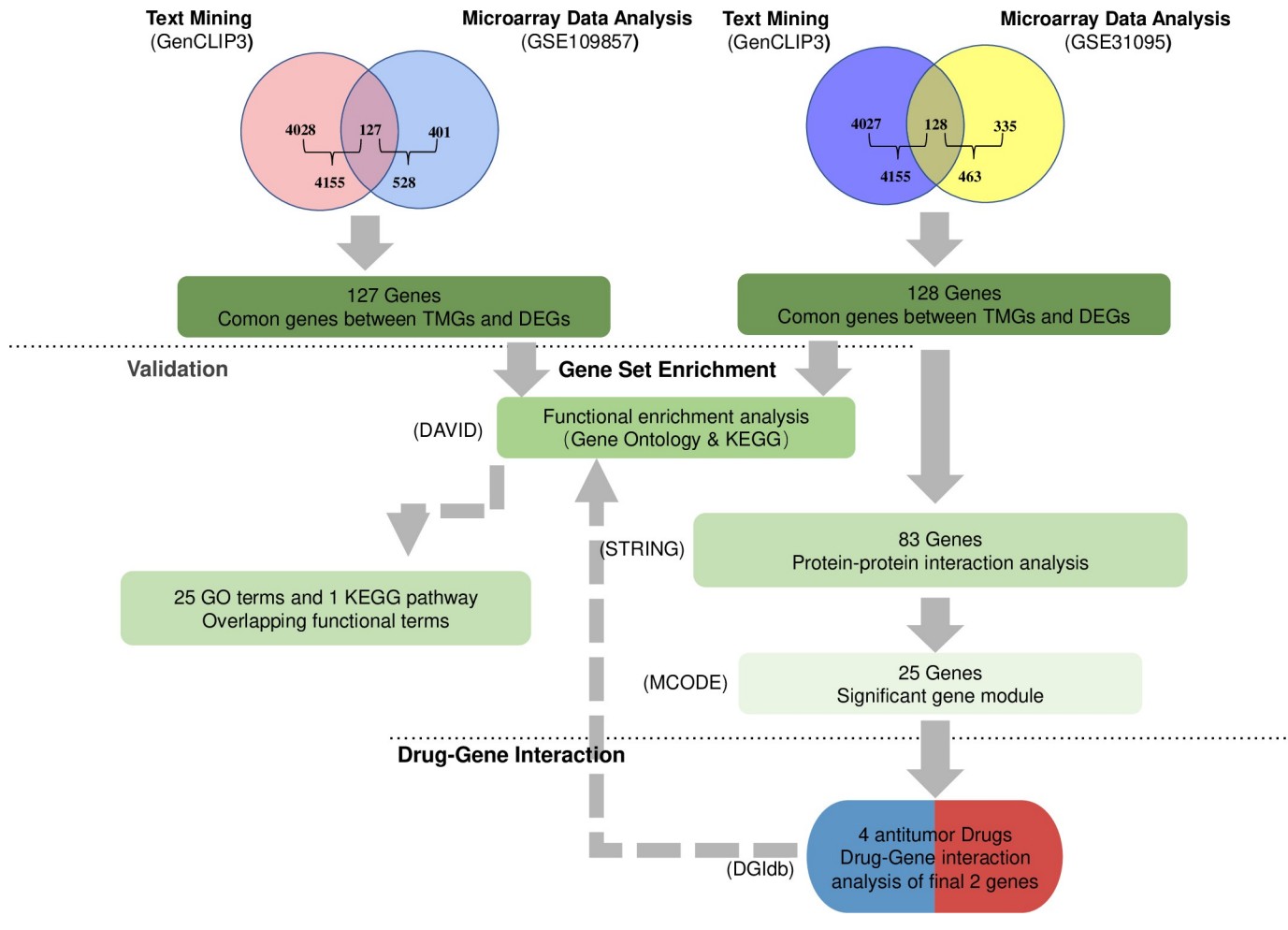

**Fig 1. The framework of data analyses.**

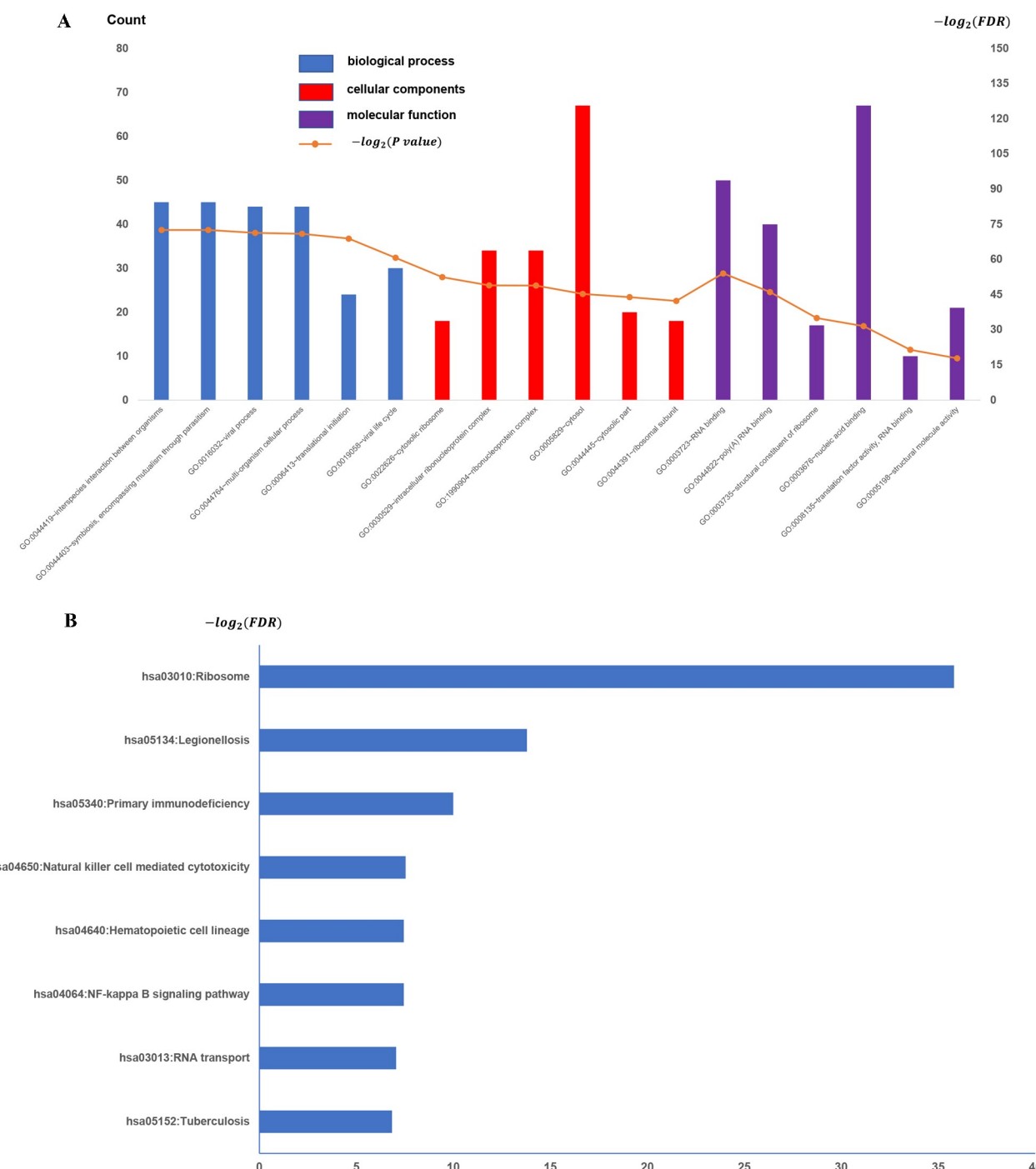

**Fig 2. All available significant gene ontology enrichment terms and signal pathway of the common genes from GSE31095 dataset.** (A) Top 10 GO terms. Number of gene of GO analysis was acquired from DAVID functional annotation tool. p <0.05. (B) KEGG pathway.

## PPI network and module analysis

The co-genes were obtained via analyzing the STRING online database (http://string-db.org) and Cytoscape software, in which 128 genes were selected to enter the PPI network complex of

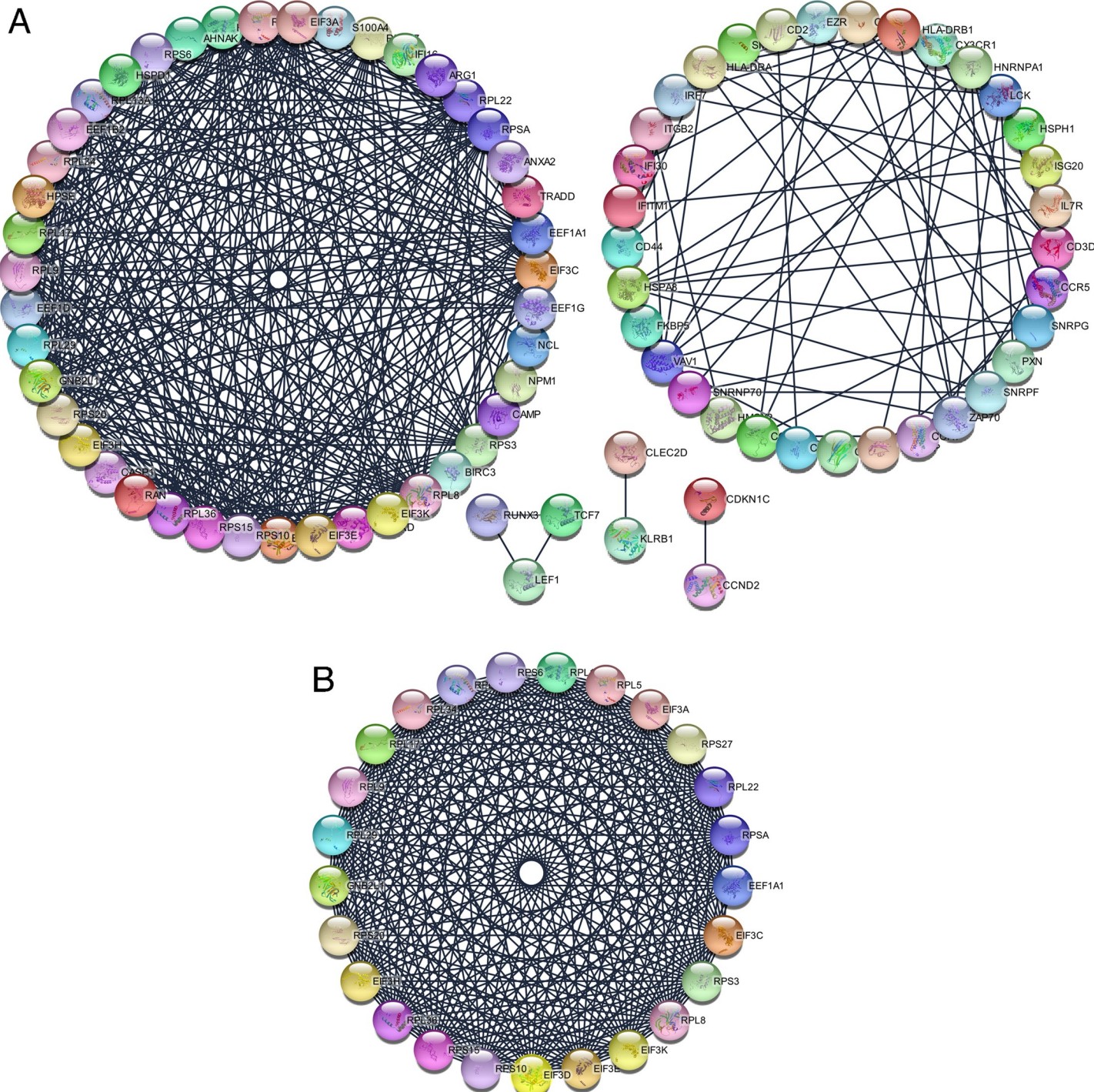

**Fig 3. The protein-protein interaction (PPI) networks construction and significant gene modules analysis.** (A) Based on the STRING online database, 128 common genes were filtered into common genes PPI network. (B) The most significant module from the PPI network.

co-genes with 83 nodes, 416 edges and a score of > 0.900 (highest confidence) (Fig 3A). Afterwards, based on MCODE, the highlighted modules were selected in the PPI network (25 nodes, 291 edges, Fig 3B).

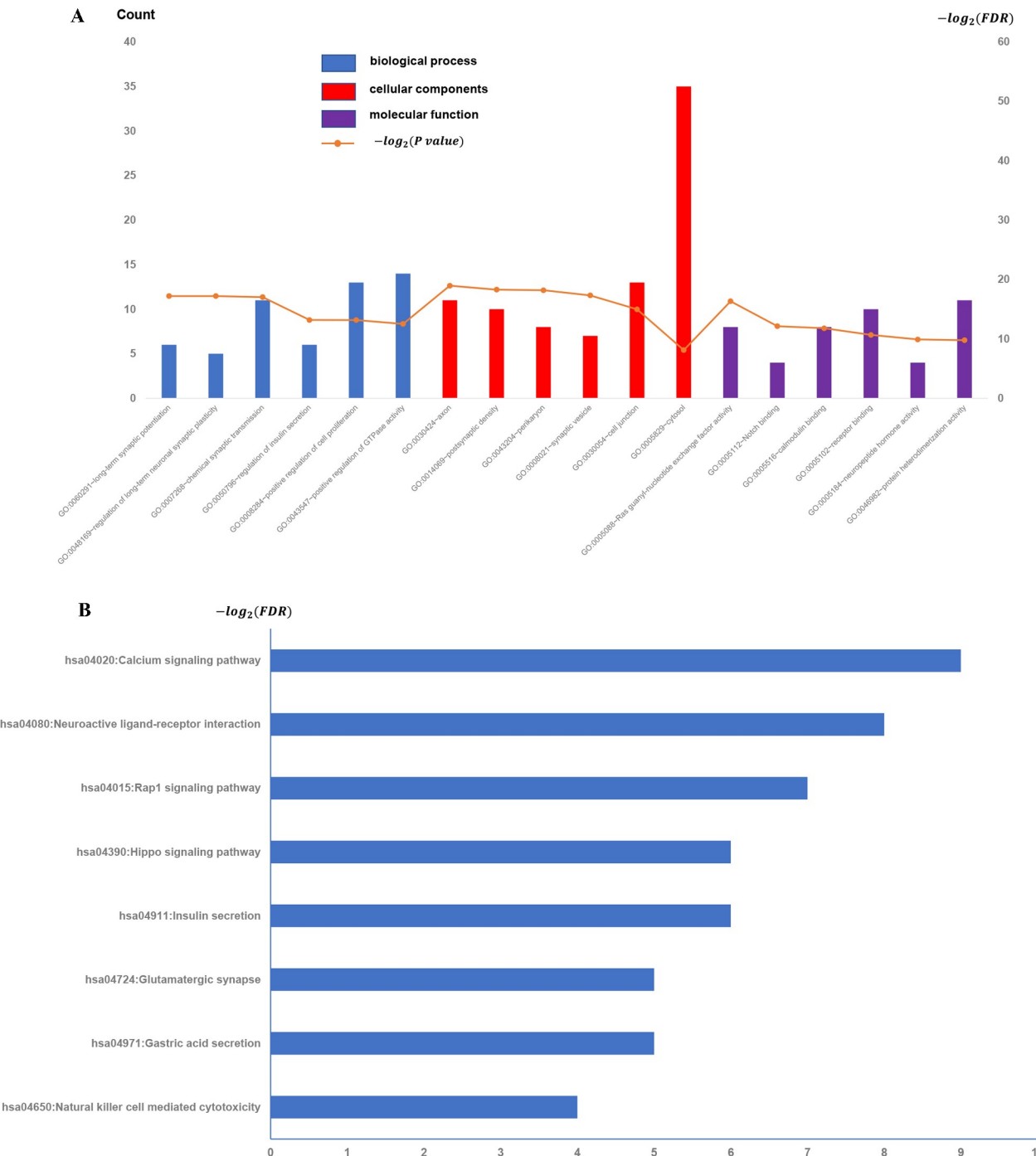

**Fig 4. All available significant gene ontology enrichment terms and signal pathway of the common genes from GSE109857 dataset.** (A) Top 10 GO terms. Number of gene of GO analysis was acquired from DAVID functional annotation tool. $p < 0.05$. (B) KEGG pathway.

### Validation in GSE109857 dataset

To test the reliability of the results derived from the GSE31095 dataset, we downloaded a cohort of five glioma samples and five normal control samples from another independent glioma dataset, GSE109857, and analyzed its gene expression data (Fig 4). Interesting, we found

**Table 1. Overlap of the enriched function terms between the two datasets.**

| Term | Category | Category |
|------|----------|----------|
| GO:0008284 | BP | positive regulation of cell proliferation |
| GO:0007417 | BP | central nervous system development |
| GO:0043065 | BP | positive regulation of apoptotic process |
| GO:0042493 | BP | response to drug |
| GO:0006468 | BP | protein phosphorylation |
| GO:0006816 | BP | calcium ion transport |
| GO:0032496 | BP | response to lipopolysaccharide |
| GO:0021537 | BP | telencephalon development |
| GO:0008285 | BP | negative regulation of cell proliferation |
| GO:0000165 | BP | MAPK cascade |
| GO:0043123 | BP | positive regulation of I-kappaB kinase/NF-kappaB signaling |
| GO:0001666 | BP | response to hypoxia |
| GO:0070374 | BP | positive regulation of ERK1 and ERK2 cascade |
| GO:0030054 | CC | cell junction |
| GO:0005576 | CC | extracellular region |
| GO:0005829 | CC | cytosol |
| GO:0043209 | CC | myelin sheath |
| GO:0009986 | CC | cell surface |
| GO:0005912 | CC | adherens junction |
| GO:0045121 | CC | membrane raft |
| GO:0005102 | MF | receptor binding |
| GO:0042802 | MF | identical protein binding |
| GO:0032403 | MF | protein complex binding |
| GO:0019899 | MF | enzyme binding |
| GO:0008092 | MF | cytoskeletal protein binding |
| hsa04650 | KEGG | Natural killer cell-mediated cytotoxicity |

GO, Gene ontology. BP. Biological processes. CC. Cellular composition. MF. Molecular function. KEGG, Kyoto Encyclopedia of Genes and Genomes.

overlap of the enriched function terms between the GSE109857 and the previous GSE31095, and it is worth noting that there are 25 GO terms in the overlapping functional terms, whereas in KEGG there is only one pathway, "Natural killer cell-mediated cytotoxicity" (Table 1).

## Drug-gene interaction and functional analysis of potential genes

Analysis of the drug-gene interaction was performed on 25 potential genes clustered in critical gene module 1. Based on the DGIdb results, there were two drugs interacted with gene EEF1A1 (eukaryotic translation elongation factor 1 alpha 1), while RPL11 (ribosomal protein L11), RPL13A (ribosomal protein L13a), RPL8 (ribosomal protein L8) and RPSA (ribosomal protein SA) were strongly associated with three different drugs, respectively. Out of these 14 drugs, only four were found to have the anti-tumor effects in glioma therapy and targeted to RPL8 and PPSA genes.

## Discussion

Glioma is a deadly malignant brain tumor with strong invasiveness, vascular hyperplasia and poor prognosis [5], and lacks of effective treatment methods. Combination therapy is

considered to be a promising approach to treat cancer for its effective anti-cancer effects and lower side effects. At present, although some progress has been made in multimodal treatment of glioma, including surgical removal, local irradiation and conventional chemotherapy [25], patients with glioma still have problems such as relapse and drug resistance, so the mortality rate of patients within two years after diagnosis is still very high [26].

In this regard, the candidate hub genes and signal pathways of glioma were screen out through a series of bioinformatics methods. 4155 genes related to Glioma were obtained through text mining and 428 DEGs were acquired by comparing the eight glioma samples with twelve normal control samples. After intersecting the set of genes obtained from text mining with the previously obtained DEGs, the common set of genes were got. Then, 25 hub genes were screened out by the network analysis of GO, KEGG and PPI. Finally, validation of our results using independent glioma dataset, GSE109857, verified that the expression of the some GO function and one KEGG pathway overlap with the previous data set (Table 1). Of these, 4 target RPL8 and RPSA and possess antineoplastic properties.

After validation through the GSE109857 dataset, the only overlapping KEGG term "Natural killer cell-mediated cytotoxicity" was obtained. Natural killer (NK) cells are essential lymphocytes that can kill virus-infected and cancer cells [27–29]. In recent studies, NK cells have been increasingly used in clinical trials in patients with cancer [30]. Studies have shown that NK cells release large amounts of interferon (IFN) -γ and are the main source of IFN - γ in the human body, and lack of NK cell-mediated production of IFN- γ is associated with an increased incidence of malignancy and infection [31].

RPL8 is reported to be involved in the occurrence of many diseases including osteosarcoma (OS) and also the corresponding treatment targets [32]. Besides, RPL8 regulates the protein synthesis process of Disc Degeneration (DD), suggesting that COL3A1 might be used for the diagnosis and treatment of DD [33]. A study of Swoboda et al. also showed that RPL8 antigen may be a relevant vaccine target for melanoma, glioma and breast cancer patients [34]. Since RPL8 is part of the ribosomal 60S subunit and participates in protein synthesis, RPL8 antigen is considered to be a relevant vaccine target for glioma [34].

Although Shi et al. recently have discovered that the RPSA gene might be related to the pyrazinamide (PZA) resistance in clinical Mycobacterium tuberculosis [35], some reports indicate that RPSA gene sequencing may not play a role in the detection of PZA sensitivity by molecular methods [36]. The correlation with tumors shows that RPSA can be used as a target for $H_2O_2$, and oxidized RPSA is found in clusters of specific adhesion molecules. In this study, we also found that RPSA oxidation *in vitro* improved the adhesion efficiency of cells to laminin [37]. Besides, RPSAs, which highly expressed in tumor cells, regulates the cell adhesion as one of its ribose *in vitro* functions and is directly related to metastatic potential [38,39]. Therefore, highly expressed RPSA in pancreatic cancer is reported to be closely related to the cancer invasion and metastasis due to the binding of RPSA-mediated cell adhesion laminin [40], further revealing a poor prognosis [41]. Another report further proved that RPSA regulates pancreatic cancer mainly through inhibiting caspase activity, which is a key protein of mediating apoptosis [42]. RPSA is also reported to be highly expressed in lung cancer, colorectal cancer, breast cancer and esophageal cancer, and RPSA can prevent tumor cells from autophagy in both breast cancer and esophageal cancer [43–45].

Four drugs (Puromycin targeting RPL8; Doxorubicin, Daunorubicin, Mitoxantrone targeting RPSA) were identified as potential drug candidates with antineoplastic activities and played the vital role in Glioma therapy.

Puromycin (RPL8), an old antibiotic derived from Streptomyces alboniger [46], is known that its antitumor activity is achieved by inhibiting 45S pre-ribosomal RNA and upstream binding factor (UBF) in MDA-MB-231 cells [47,48].It also has been found to induce apoptosis

in breast cancer cells by insulin-like growth factor 1 (IGF-I), because it prevents the ribosomal protein generate process by causing the premature release of a polypeptide from the ribosome in malignant cells. In addition, studies have proved that puromycin can enhance its antineoplastic effect via combining with other drugs, such as RITA or doxorubicin, which can be effectively used for wild-type P53 cancers [49].

Daunorubicin (RPSA) is a functional drug that exerts the antineoplastic effects through direct cytotoxicity and an apoptosis-inducing effect through the apoptotic signaling pathways in the cell cytoplasm and mitochondria. As a chemotherapy strategy for treating brain glioma, functionally targeted daunorubicin liposomes not only have the ability to eliminate gliomas, but also have the potential to remove glioma stem cells [50]. Meanwhile, the double-targeted daunorubicin liposomes can improve the therapeutic effect of glioma both *in vitro* and *in vivo*, and also significantly increase the transport rate of the blood-brain barrier model, up to 24.9%.

Doxorubicin (DOX) is identified as one of the most common and economic chemotherapy drugs in the treatment of malignant gliomas. However, when DOX is used alone, its clinical application is limited by its serious side effects [51,52]. Therefore, many drugs that could be combined with DOX are found in a series of subsequent studies. Among them, Gao et al., constructed a novel combination therapy to synthesize 131I-DOX-NL using two traditional drugs, DOX and 131I, which not only significantly reduced the side effects of DOX, but also effectively played an antitumor effect [53]. Besides, doxorubicin combined with dacarbazine is often used as a first-line treatment for leiomyosarcoma [54–60].

As mentioned above, the most common treatment for cancer is combination therapy [61,62], as is MTO. In previous studies, MTO has been found to be extensively used to treat metastatic, and castration-resistant prostate cancer, acute myeloid and lymphoblastic leukemias [63–68].

Up to date, the genes and drugs we have identified are only preliminarily studied in previous studies. Therefore, if further verification of its accuracy is needed, the above results need to be combined with basic experiments or computer simulations. In recent years, Chen's professional research team has developed a computer model of miRNA-disease association prediction (MDHGI) to discover new miRNA-disease associations by integrating the predicted association probability obtained from matrix decomposition through sparse learning method [69–72]. If this model is included in biometric analysis, a broader simulation can be carried out through big data and disease data can be accurately analyzed, so as to obtain more targeted genes and targeted therapy drugs for future clinical research and treatment.

## Conclusions

In summary, we analyzed a GSE31095 dataset and performed functional enrichment analysis. We then validated our approach on an independent GSE109857 dataset. Finally, 2 identified potential genes (RPL8 and RPSA) were analyzed on DGIdb and four potential antitumor drugs (Puromycin, Doxorubicin, Daunorubicin and Mitoxantrone) identified. Some of the identified genes are potential glioma biomarkers. Characterization of the identified drugs will offer more insights into potential, novel therapeutic strategies against glioma.

## Acknowledgments

Thanks to Bin Zhao (Official Wechat Account: SCIPhD) of ShengXinZhuShou for English editing on the manuscript.

## Author Contributions

**Conceptualization:** Zhengye Jiang.

**Data curation:** Yanxi Shi.

**Methodology:** Guowei Tan.

**Supervision:** Zhanxiang Wang.

**Writing – original draft:** Zhengye Jiang.

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
