## [Decision Letter · Decision Letter 0]

12 Nov 2020

PONE-D-20-26541

Identification of Glioma-Related Potential Genes and Drugs: Based on GEO Database and Mining text

PLOS ONE

Dear Dr. Wang,

Thank you for submitting your manuscript to PLOS ONE. After careful consideration, we feel that it has merit but does not fully meet PLOS ONE’s publication criteria as it currently stands. Therefore, we invite you to submit a revised version of the manuscript that addresses the points raised during the review process.

We look forward to receiving your revised manuscript.

Kind regards,

Edwin Wang

Academic Editor

PLOS ONE

Additional editor comments:

1) Please clarify in the GO/KEGG analysis the p values were FDR-adjusted

2) Please clarify if statistical measures have been used in the drug-gene analysis

Journal Requirements:

3. Please ensure that all steps undertaken in the data collection and analysis are included in the Methods section, in enough detail for another researcher to replicate the analysis.

5. Please include your tables as part of your main manuscript and remove the individual files.

Please note that supplementary tables should be uploaded as separate "supporting information" files.

Reviewers' comments:

Reviewer's Responses to Questions

**Comments to the Author**

1. Is the manuscript technically sound, and do the data support the conclusions?

Reviewer #1: Partly

Reviewer #2: Yes

2. Has the statistical analysis been performed appropriately and rigorously? 

Reviewer #1: No

Reviewer #2: Yes

3. Have the authors made all data underlying the findings in their manuscript fully available?

Reviewer #1: Yes

Reviewer #2: Yes

4. Is the manuscript presented in an intelligible fashion and written in standard English?

Reviewer #1: No

Reviewer #2: No

5. Review Comments to the Author

Reviewer #1: According to the policy of PLoS One, here I will only comment about the technical issues of the manuscript:

1. There are several glioma gene expression datasets in GEO database. Why did the authors consider only GSE31095, which is a relatively old, small-size microarray dataset in the analysis? Considering consensus DEGs in multiple gene expression datasets will prominently improve the confidence of the resulted gene list.

2. As for the drug analysis, the authors picked the drugs targeting hub genes (which were mostly known cancer driver genes) as the potential glioma-related drugs. The authors can implement gene expression-based drug analysis like Connectivity Map analysis to validate the predicted drugs.

3. The threshold of statistical significance was vague: P-value, adjusted P-value, or FDR, which one was used should be consistent.

4. Grammatical errors, awkward expressions or typos could be observed in nearly everyy paragraph of the manuscript. For example, the title should be “Computational Screening of Potential Glioma-Related Genes and Drugs Based on Analysis of GEO Dataset and Text Mining” rather than “Identification of Glioma-Related Potential Genes and Drugs: Based on GEO Database and Mining text”. Careful language editing by a native speaker is necessary before further consideration of the manuscript.

Reviewer #2: The authors identified some potential Glioma-related genes and available drugs based on analyzing GEO datasets and mining text. I hope the manuscript could be further strengthened by the following comments.

1. Please clearly state the major innovation of this work.

2. I want to know whether the researches for verifying your identification (the researches of Swoboda et al. and Shi et al.) are included in the GenCLIP3 platform for text mining.

3. I want to know whether this method can also be used for the analysis of other cancer.

4. You should revise your English writing carefully and eliminate small errors in the paper to make the paper easier to understand.

5. Could you give some discussions whether your method could be used to predict glioma-related potential non-coding RNAs as the future direction of this work (PMIDs: 29939227, 29045685, 30142158, and 27345524)?

6. PLOS authors have the option to publish the peer review history of their article (what does this mean?). If published, this will include your full peer review and any attached files.

Reviewer #1: No

Reviewer #2: No

---

## [Author Response · Author response to Decision Letter 0]

7 Dec 2020

Dear Editor Edwin Wang and Reviewers:

We are very grateful to Reviewer for reviewing the paper so carefully. We have carefully considered the suggestion of Reviewer and make some changes.

Responds to the Editor Wang’s comments:

1 Please clarify in the GO/KEGG analysis the p values were FDR-adjusted.

Answer: We appreciate and thank for the detailed Editor Wang of our manuscript. We are very sorry for our negligence of the explanation and we have adjusted the P values by FDR in GO and KEGG analyses.

2 Please clarify if statistical measures have been used in the drug-gene analysis.

Answer: The source of the drug score in the drug-gene database is based on its source. For example, one score is derived from published articles while another score is derived from the database. This is a descriptive conclusion without using statistical method. Thank you very much for your great efforts on our manuscript. 

Responds to the reviewers' comments:

Reviewer #1：

1. There are several glioma gene expression datasets in GEO database. Why did the authors consider only GSE31095, which is a relatively old, small-size microarray dataset in the analysis? Considering consensus DEGs in multiple gene expression datasets will prominently improve the confidence of the resulted gene list.

Answer: We appreciate it very much for this good suggestion. However, this data set has been already used by our research team in relevant exploration and mining, so I further analyzed and mined it with text mining through generating letters. Coincidentally, I got several genes that I was studying, such as RPL8 and RPSA.

2. As for the drug analysis, the authors picked the drugs targeting hub genes (which were mostly known cancer driver genes) as the potential glioma-related drugs. The authors can implement gene expression-based drug analysis like Connectivity Map analysis to validate the predicted drugs.

Answer: We would like to thank the reviewer fort his comment. As reviewers have pointed out, the drugs selected are indeed those with known oncogenes, but they are not included in the treatment guidelines for gliomas. Therefore, in this study, we screened out these potential tumor drugs to lay a foundation for the following basic experiments, hoping to expand the indications of these drug therapy and provide new possibilities for the targeted therapy of glioma in the future. 

3. The threshold of statistical significance was vague: P-value, adjusted P-value, or FDR, which one was used should be consistent.

Answer: We are very sorry for our negligence of the explanation. We have already adjusted the statistical thresholds to FDR/ ad-value (tip: FDR and ad-value are the same thing). We would like to thank the reviewer also fort his comment.

4. Grammatical errors, awkward expressions or typos could be observed in nearly everyy paragraph of the manuscript. For example, the title should be “Computational Screening of Potential Glioma-Related Genes and Drugs Based on Analysis of GEO Dataset and Text Mining” rather than “Identification of Glioma-Related Potential Genes and Drugs: Based on GEO Database and Mining text”. Careful language editing by a native speaker is necessary before further consideration of the manuscript.

Answer: We apologize for the poor language of our manuscript. The manuscript has been revised by a native English speaker for language corrections. We really hope that the flow and language level have been substantially improved. Many thanks go to reviewers, and we are feel so warm for your suggestions.

Reviewer #2：

1. Please clearly state the major innovation of this work.

Answer: We thank the reviewer. The innovation of our research lies in the cross-combination of data sets in GEO database and text mining. Through the bioinformatics analysis, differential genes are screened out and potential targeted drugs are further explored through differential genes, which will provide new targets and indications for the clinical treatment of glioma.

2. I want to know whether the researches for verifying your identification (the researches of Swoboda et al. and Shi et al.) are included in the GenCLIP3 platform for text mining.

Answer: Dear reviewer, I think you may have misunderstood. The research of Swoboda et al. and Shi et al. is only used to prove that the differentially expressed genes I have obtained are related to other tumors, and these differentially expressed genes are derived from the GenCLIP3 platform and the GSE31095 data set. Therefore, only differentially expressed genes are included in the GenCLIP3 platform. We appreciate and thank for the detailed review of our manuscript.

3. I want to know whether this method can also be used for the analysis of other cancer.

Answer: We would like to thank the reviewer fort his comment. I think it can be used for analyzing many cancers. Because bioscientific research is based on big data survey research, this also means that as long as there is reliable and sufficient tumor sample data, it can be fully applied to other research. Therefore, our current glioma research is not a special case, but a microcosm of research directions.

4. You should revise your English writing carefully and eliminate small errors in the paper to make the paper easier to understand.

Answer: We apologize for the poor language of our manuscript. The manuscript has been revised by a native English speaker for language corrections. We really hope that the flow and language level have been substantially improved. We would like to thank the reviewer also fort his comment.

5. Could you give some discussions whether your method could be used to predict glioma-related potential non-coding RNAs as the future direction of this work (PMIDs: 29939227, 29045685, 30142158, and 27345524)?

Answer: Thank you for your valuable advice. First of all, what I want to say is that non-coding RNA has a very large application prospect. This also means that it can be studied by the method of biosynthesis in the research of glioma. Similarly, the research of biosynthesis is only the initial exploratory research, so if you need to strengthen the reliability, it must be combined with other methods to increase its credibility. The fundamental experiment cycle is too long, and the current popular computer models are just in line. Just like the 4 articles mentioned by the reviewer, after reading carefully, the author found that these articles mainly describe a professional computer model for MiRNA–Disease Association prediction (IMCMDA). This is a surprising discovery, if the big data of Bioinformatics analysis is combined with this model, it can greatly improve the prediction of miRNAs for diseases, not only for gliomas, but for any other tumors, or even any other diseases. This will be an innovation in the era of computer big data, and the author will further explore the correlation between the two. At the same time, we have quoted these 4 articles on Model for MiRNA–Disease Association prediction (IMCMDA) into this article, and have discussed them in the discussion part of this research, in order to make this combined method familiar and understood by more people. Many thanks go to reviewers, and we are feel so warm for your suggestions.

---

## [Decision Letter · Decision Letter 1]

4 Jan 2021

PONE-D-20-26541R1

Computational Screening of Potential Glioma-Related Genes and Drugs Based on Analysis of GEO Dataset and Text Mining

PLOS ONE

Dear Dr. Wang,

Thank you for submitting your manuscript to PLOS ONE. After careful consideration, we feel that it has merit but does not fully meet PLOS ONE’s publication criteria as it currently stands. Therefore, we invite you to submit a revised version of the manuscript that addresses the points raised during the review process.

We look forward to receiving your revised manuscript.

Kind regards,

Edwin Wang

Academic Editor

PLOS ONE

Reviewers' comments:

Reviewer's Responses to Questions

**Comments to the Author**

1. If the authors have adequately addressed your comments raised in a previous round of review and you feel that this manuscript is now acceptable for publication, you may indicate that here to bypass the “Comments to the Author” section, enter your conflict of interest statement in the “Confidential to Editor” section, and submit your "Accept" recommendation.

Reviewer #1: (No Response)

Reviewer #2: (No Response)

2. Is the manuscript technically sound, and do the data support the conclusions?

Reviewer #1: Partly

Reviewer #2: Yes

3. Has the statistical analysis been performed appropriately and rigorously? 

Reviewer #1: Yes

Reviewer #2: Yes

4. Have the authors made all data underlying the findings in their manuscript fully available?

Reviewer #1: Yes

Reviewer #2: Yes

5. Is the manuscript presented in an intelligible fashion and written in standard English?

Reviewer #1: (No Response)

Reviewer #2: Yes

6. Review Comments to the Author

Reviewer #1: The authors have addressed or explained most of my previous points except Major point 1. Without supporting evidence from another gene expression dataset, the technical quality requirement, which is emphasized by PLoS One journal, could not be met. On the other hand, this is not a hard task: The authors can simply find another glioma RNA-Seq dataset from GEO, intersect its differentially expressed genes with GenCLIP3 gene set, and perform GO and KEGG functional enrichment analysis. If there are some overlap of the enriched function terms between the new and the previous analyses, the result of this manuscript should be much more consolidated.

Besides, I would also like to point out that adjusted p-value and FDR are actually NOT the same thing. There are several statistical methods for p-value adjustment against multiple tests; and FDR, often following the Benjamini family of methods, is one category of adjusted p-value.

Reviewer #2: Authors should carefully check the information of references. For example, the author of [64] should be Chen X, Wang L, Qu J, Guan N-N, Li J-Q and its volume, issue and page number should be 34(24): 4256-4265; the year, volume, issue and page number of [66] should be 2017 18(4):558-576; the author of [67] should be Chen X, Yin J, Qu J, Huang L and its page number should be e1006418.

7. PLOS authors have the option to publish the peer review history of their article (what does this mean?). If published, this will include your full peer review and any attached files.

Reviewer #1: No

Reviewer #2: No

---

## [Author Response · Author response to Decision Letter 1]

21 Jan 2021

Responds to the reviewers' comments:

Reviewer #1：

1. The authors have addressed or explained most of my previous points except Major point 1. Without supporting evidence from another gene expression dataset, the technical quality requirement, which is emphasized by PLoS One journal, could not be met. On the other hand, this is not a hard task: The authors can simply find another glioma RNA-Seq dataset from GEO, intersect its differentially expressed genes with GenCLIP3 gene set, and perform GO and KEGG functional enrichment analysis. If there are some overlap of the enriched function terms between the new and the previous analyses, the result of this manuscript should be much more consolidated.

Besides, I would also like to point out that adjusted p-value and FDR are actually NOT the same thing. There are several statistical methods for p-value adjustment against multiple tests; and FDR, often following the Benjamini family of methods, is one category of adjusted p-value.

Answer: We would like to thank the reviewer fort his comment. As reviewers have pointed out, without supporting evidence from another gene expression dataset, the technical quality requirement could not be met. In response to this, we followed the reviewer’s recommendations and methods, after continuous mining and exploration of other datasets in GEO, repeated analysis and verification, and finally found a dataset GSE109857. Through verification, some overlap of the enriched function terms between the new and the previous analyses, the final results have been listed and modified in the article.

Secondly, follow the reviewer’s description of both the FDR and the adjusted P-value, we have repeatedly checked the literature and found that, as the reviewer said, the two are not the same thing. Thank the reviewers for such important and valuable comments. And the threshold used in this article is FDR.

Reviewer #2：

1. Authors should carefully check the information of references. For example, the author of [64] should be Chen X, Wang L, Qu J, Guan N-N, Li J-Q and its volume, issue and page number should be 34(24): 4256-4265; the year, volume, issue and page number of [66] should be 2017 18(4):558-576; the author of [67] should be Chen X, Yin J, Qu J, Huang L and its page number should be e1006418.

Answer: We appreciate the reviewer for pointing out this fact. We have checked the references thoroughly are now all in a uniform format in the revised manuscript.

---

## [Editor Report · Decision Letter 2]

10 Feb 2021

Computational Screening of Potential Glioma-Related Genes and Drugs Based on Analysis of GEO Dataset and Text Mining

PONE-D-20-26541R2

Dear Dr. Wang,

We’re pleased to inform you that your manuscript has been judged scientifically suitable for publication and will be formally accepted for publication once it meets all outstanding technical requirements.

Kind regards,

Edwin Wang

Academic Editor

PLOS ONE
---

## [Editor Report · Acceptance letter]

17 Feb 2021

PONE-D-20-26541R2 

Computational Screening of Potential Glioma-Related Genes and Drugs Based on Analysis of GEO Dataset and Text Mining 

Dear Dr. Wang:

I'm pleased to inform you that your manuscript has been deemed suitable for publication in PLOS ONE. Congratulations! Your manuscript is now with our production department. 

Kind regards, 

on behalf of

Dr. Edwin Wang 

Academic Editor

PLOS ONE